# Absolute risk-based versus individualized benefit approaches for determining statin eligibility in primary prevention of cardiovascular diseases in Chinese populations: A modeling study

Qiuping Liu[1], Chao Gong[1], Tianjing Zhou[1], Minglu Zhang[1], Xiaofei Liu[1], Xun Tang[1,2,3]ᴥ*, Pei Gao[1,2,3]ᴥ*

1 Department of Epidemiology and Biostatistics, Peking University School of Public Health, Beijing, China,
2 Center for Real-world Evidence evaluation, Peking University Clinical Research Institute, Beijing, China,
3 Key Laboratory of Epidemiology of Major Disease, Peking University, Ministry of Education, Beijing, China

ᴥ These authors contributed equally to the work.
* tangxun@bjmu.edu.cn (XT); peigao@bjmu.edu.cn (PG)

## Abstract

### Background

Current guidelines for statin use in primary prevention of cardiovascular disease (CVD) predominantly rely on absolute 10-year CVD risk scores. However, this approach may not adequately capture heterogeneity in the potential benefit of low-density lipoprotein cholesterol (LDL-C) reduction. This study compares the absolute risk-based approach with an individualized benefit approach, based on the Causal-Benefit model considering predicted lipid-lowering effects, for statin eligibility in Chinese populations.

### Methods and Findings

We analyzed nationally representative data from the China Health and Retirement Longitudinal Study, including adults aged 40–80 years, free of diabetes and CVD history, with LDL-C levels between 1.8 mmol/L and 4.9 mmol/L, and no prior statin use. Statin eligibility was determined using two strategies: (i) the absolute risk-based approach (10-year CVD risk), and (ii) the individualized benefit approach (using the Causal-Benefit model framework incorporating predicted individual absolute risk reduction [iARR]). We estimated eligible populations, CVD events averted, and number needed to treat (NNT) both at population and individual level (iNNT) over 10 years versus no treatment, assessed discordance, and primarily calibrated the benefit threshold to match event prevention by the risk-based approach for comparison. A total of 7,287 adults were analyzed, forming a cohort reflective of 324.6 million Chinese adults (mean age 57 years; 51.7% women). To prevent a similar number

**Data availability statement:** The CHARLS is a public database; the data are available via the website https://charls.charlsdata.com/pages/Data/2015-charls-wave4/en.html. The code used in the analysis was adopted from the Prevention Impact and Efficiency (PIE) model: https://github.com/markpletcher/PIE-Model_Stata.

**Funding:** This study was supported by the Noncommunicable Chronic Diseases - National Science and Technology Major Project (2024ZD0527406 to X.T., https://service.most.gov.cn/), the National Natural Science Foundation of China (82373662, 81973132 to X.T., https://www.nsfc.gov.cn/english/site_1/index.html), and the Beijing Natural Science Foundation (IS24047 to P.G., https://nsf.kw.beijing.gov.cn/bjnsfweb/). The study funders/sponsors had no role in the study design, data collection and analysis, decision to publish, or preparation of the manuscript.

**Competing interests:** The authors have read the journal's policy and declare the following competing interests: P.G. reported receiving research funds from Bayer and Merck; however, the funding sources had no relation to this study. All other authors of this study have reported that they have no relationships relevant to the contents of this paper to disclose.

**Abbreviations:** ARR, absolute risk reduction; CHARLS, China Health and Retirement Longitudinal Study; CI, confidence interval; CVD, cardiovascular disease; GATHER, Guidelines for Accurate and Transparent Health Estimates Reporting; iARR, individual absolute risk reduction; IQR, interquartile range; LDL-C, low-density lipoprotein cholesterol; LMICs, low- and middle-income countries; NNT, number needed to treat; PIE, Prevention Impact and Efficiency; WHO, World Health Organization.

of CVD events (2.19 million *vs.* 2.16 million), 49.2 million (95% confidence interval [CI]: 45.3,53.0) and 50.3 million (95% CI: 46.0,54.6) adults would be eligible for statins therapy under the individualized benefit and absolute risk-based approaches, respectively. Among 58.9 million adults eligible for either strategy, the concordance was only 68.9%. The benefit approach alone identified 8.6 million people highly benefit from statin therapy, who would not be eligible for statin therapy under the absolute risk-based approach, and this includes 1.3 million people with borderline risk (5% to 7.5%). Conversely, the risk-based approach selected more individuals with low predicted benefit (minimum iARR: 2.5% *vs.* 3.4%), resulting in a less efficient individual-level targeting profile (maximum iNNT: 41 *vs.* 29). A key limitation of this study is that benefit was estimated primary from LDL-C reduction, which may neglect other biological mechanisms of statin effects and underestimate the total benefit.

## Conclusions

The individualized benefit approach prioritizes individuals most likely to benefit from statin therapy, differing from conventional risk-based selection through its superior individual-level precision. This approach can enhance the capacity to discriminate treatment effects at the individual level, making it particularly valuable for shared decision-making in resource-constrained settings.

## Author summary

### Why was this study done?

- Current guidelines relying on 10-year absolute risk might recommend statins for some high-risk individuals who gain limited benefit specifically from LDL-C reduction, while potentially missing lower-risk individuals who stand to gain substantially more benefit from LDL-C lowering.

- An alternative, the individualized benefit approach (conceptually based on the Causal-Benefit model), estimates benefit based on modifiable causal factors like LDL-C and might better identify individuals, including younger people with high LDL-C, who could derive substantial benefit.

- The potential population-level implications of the individualized benefit approach needed quantification in settings like China, where non-lipid factors (e.g., hypertension) are key contributors to overall risk.

### What did the researchers do and find?

- Using nationally representative Chinese data, we compared statin eligibility recommendations between the conventional absolute risk-based and individualized benefit approaches.

- When calibrated to prevent a similar number of cardiovascular events, the groups recommended for treatment overlapped by only about 69%. The group selected by the individualized benefit approach contained a higher concentration of individuals predicted to receive substantial absolute benefit (iARR) from statins, suggesting this method better targets those likely to benefit most.

- Applying the Causal-Benefit model's minimum benefit threshold principle, the benefit approach recommended treatment for more adults (46 million additional) compared to the high-risk approach alone, increasing projected 10-year CVD events prevented by 65% (estimated 3.57 million vs. 2.16 million).

## What do these findings mean?

- In resource-constrained settings, the individualized benefit approach may better target statins to those likely to benefit most, potentially avoiding overtreatment in some high-risk individuals and allowing earlier treatment for younger populations with high LDL-C.

- A key study limitation is the primary focus on LDL-C reduction for benefit estimation, which neglects other biological mechanisms of statin effects that might underestimate the total benefit, especially for individuals deemed high-risk for non-lipid reasons. However, in settings with sufficient resources that allow for shared decision-making, using a minimum benefit threshold (as in the Causal-Benefit model framework) could identify a broader group that potentially benefits from statins through various mechanisms.

## Introduction

Cardiovascular disease (CVD) primary prevention guidelines typically recommend statins to individuals whose predicted 10-year absolute risk surpasses a certain threshold [1–5]. Although this approach is widely accepted, it can overlook meaningful differences in how much individuals might actually benefit from low-density lipoprotein cholesterol (LDL-C) reduction [6,7]. This limitation is particularly evident in populations from low- and middle-income countries (LMICs) such as China, where non-lipid risk factors, rather than markedly high LDL-C, drive absolute CVD risk [8,9]. Consequently, individuals identified as "high-risk" by conventional models may be recommended for treatment despite achieving limited LDL-C reductions, while those with moderately lower absolute risk but higher LDL-C levels may be excluded from treatment initiation despite their potential for substantial absolute risk reduction (ARR) [10].

Some recent guidelines suggest broader statin consideration, reflecting a move towards more personalized prevention [1]. An individualized benefit approach, conceptually based on the Causal-Benefit model framework [10,11], formalizes this by evaluating the predicted individual absolute risk reduction (iARR) from LDL-C lowering. Estimating such individualized treatment effects aims to enable more precise and efficient interventions, even with established therapies like statins. Prioritizing individuals predicted to achieve greater iARR may improve targeting compared to selecting solely based on high absolute risk, particularly by avoiding treatment expansion to high-risk individuals unlikely to derive substantial benefit specifically from LDL-C reduction. Although conceptually appealing, this individualized benefit approach is not standard practice, and its population-level implications require evaluation, particularly regarding efficiency and population characteristics in LMICs with distinct risk profiles and healthcare contexts.

Therefore, this study compares the conventional absolute risk-based approach with an individualized benefit approach in a nationwide representative sample of Chinese adults (aged 40–80 years, without prevalent CVD or diabetes, with baseline LDL-C 1.8–4.9 mmol/L, and no prior statin use). Specifically, the study seeks to (1) quantify differences in statin eligibility and predicted CVD events prevented between the two approaches at the population level; (2) evaluate reclassification patterns, particularly the prioritization of individuals with higher predicted benefit from LDL-C lowering under

the individualized benefit approach; and (3) explore the potential clinical implications of using an individualized benefit approach to guide patient-clinician shared decision-making in primary prevention.

## Methods

### Ethics statement

Data were drawn from the China Health and Retirement Longitudinal Study (CHARLS), a nationally representative longitudinal survey of Chinese adults using a multi-stage stratified probability-proportionate-to-size sampling method [12,13]. Ethical approval for the CHARLS study was granted by the Institutional Review Board at Peking University (IRB00001052-11015). The blood-based biomarker collection component of the study was also separately approved by the Institutional Review Board at Peking University (IRB00001052-11014) [14]. All participants gave written informed consent before participation.

### Data sources and study population

The anonymized data are publicly available on the CHARLS website (https://charls.charlsdata.com/pages/Data/2015-charls-wave4/en.html). Five waves of data have been collected, with the 2015 wave being the latest available dataset containing blood samples, including lipid information. In the current study, based on the blood sample of the 2015 wave, we included participants aged 40–80 years in 2015, excluding those with a history of CVD, diabetes, LDL-C level lower than 1.8 mmol/L, LDL-C level greater than or equal to 4.9 mmol/L, and statin use before 2015 (S1 Fig). Demographic data from the baseline (2011), wave 2 (2013), wave 4 (2018), and wave 5 (2020) were used to supplement age and sex information for participants in the 2015 wave, while disease history data were based on information available before 2015.

### Strategies of eligibility for statin therapy

Statin eligibility was assessed using two strategies: (i) the conventional absolute risk-based approach and (ii) the individualized benefit approach. Both required estimating baseline 10-year CVD risk, and the latter required the expected iARR from statin treatment (Conceptual overview in Fig 1, details are provided in the S1 Text).

For the absolute risk-based approach, we calculated 10-year absolute CVD risk using the 2019 World Health Organization (WHO) laboratory-based equations, incorporating age, sex, smoking status, systolic blood pressure, and total cholesterol [15]. Based on published validation data indicating overestimation in Chinese populations [16], we recalibrated WHO risk estimates by dividing predicted risk by 1.59 for men and 1.72 for women. Consistent with Chinese-specific recommendations [17] and aligning with widely used categories [5], we defined risk strata as high (≥10%), intermediate (7.5% to <10%), borderline (5% to <7.5%), and low (<5%).

The individualized benefit approach, conceptually based on the Causal-Benefit model [11], guided eligibility by predicted iARR from moderate-intensity statin therapy (S1 Text). Consistent with this framework and established methods for estimating individualized benefit [10,11], we assumed LDL-C reduction is the primary mediator of statin benefit for calculating iARR (details in S1 Text). The estimation involved sequentially calculating: (i) untreated 10-year risk (using the recalibrated WHO equations); (ii) expected absolute LDL-C reduction (assuming a standard 40% reduction from baseline LDL-C for moderate-intensity therapy [18]); (iii) the relative risk (RR) of CVD per mmol/L of LDL-C reduction, using methods that account for potential interaction with baseline risk [10]; (iv) the overall RR of CVD with statin treatment derived from (ii) and (iii); (v) the predicted on-treatment 10-year risk (calculated as untreated risk multiplied by the overall RR with treatment); and finally (vi) the iARR (calculated as untreated risk minus on-treatment risk). Multiple potential iARR thresholds were evaluated as described below.

We applied two main principles for setting iARR thresholds: (1) Comparable Event Prevention: For direct comparison with the risk-based approach, we determined iARR thresholds ("high-benefit" and "moderate-benefit") by calibrating them to yield a predicted number of 10-year prevented CVD events equivalent to that achieved by the absolute risk-based

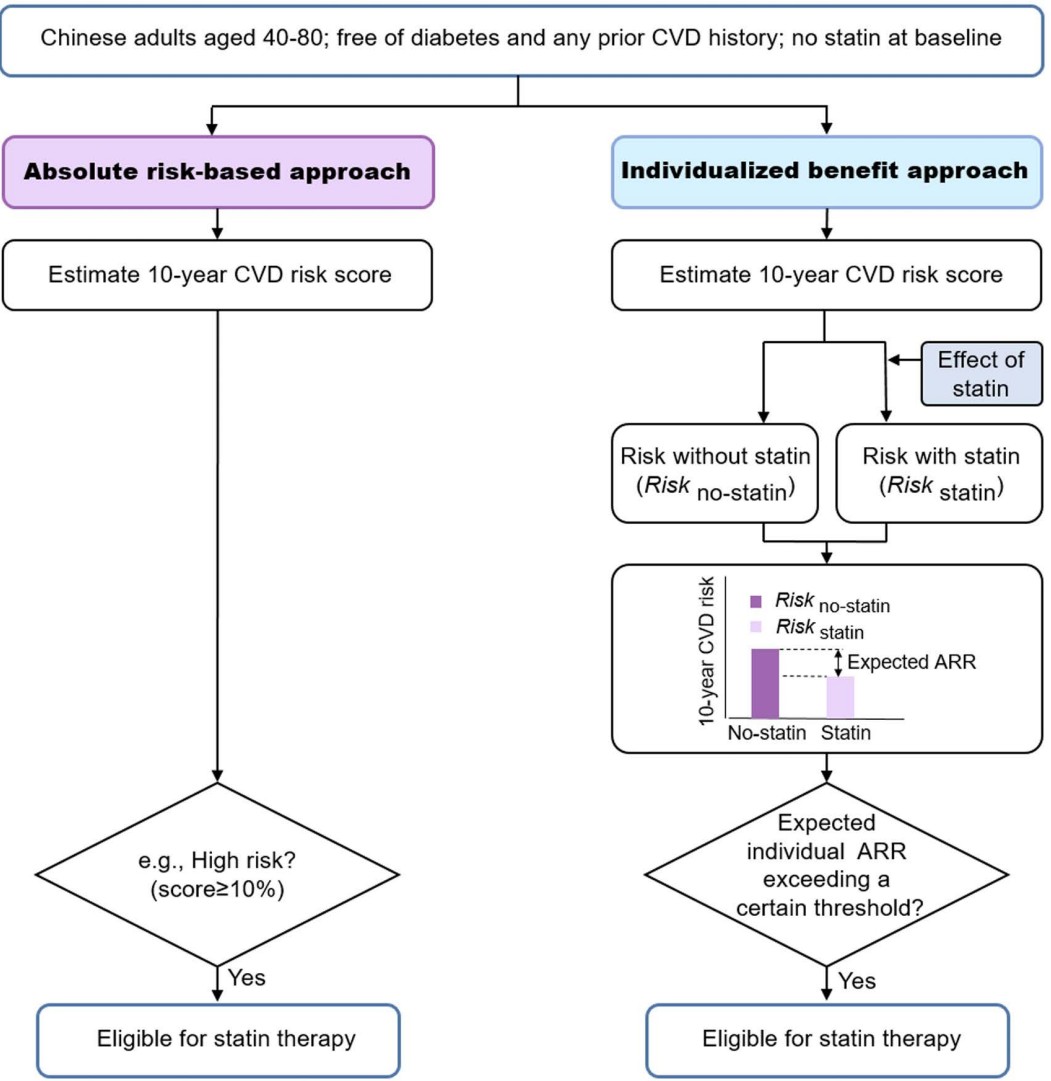

**Fig 1. Conceptual diagram of different approaches for statin eligibility.** The effect of statin was accounted for the potential benefits of lipid-lowering, calculated by $RR_{per unit of LDL-C reduction}^{(absolute LDL-C reduction)}$. LDL-C indicates low-density lipoprotein cholesterol; RR, relative risk; iARR, individual absolute risk reduction.

approach using high-risk (≥10%) and intermediate-or-higher risk (≥7.5%) cut-offs, respectively. This allowed a direct comparison of population characteristics and efficiency under matched population-level numbers of CVD events prevented. (2) Minimum Benefit Expansion (Causal-Benefit Principle): To align with the Causal-Benefit model's typical application, we also performed analyses using iARR thresholds defined by the minimum predicted iARR observed within the high-risk (≥10%) and intermediate-or-higher-risk (≥7.5%) groups identified by the absolute risk-based approach.

## Statistical analysis

Baseline characteristics are presented as means (SD) for continuous variables (except for baseline absolute risk and expected risk reduction), medians with interquartile ranges (IQRs) for baseline absolute risk and iARR, and percentages

for categorical variables. Intergroup comparisons for continuous and categorical variables were performed using *t*-tests and Chi-squared tests. All tests were two-sided, with a significance threshold of 0.05.

We projected the 10-year CVD events averted, number and proportion eligible for statins, and average NNT versus no treatment using the code of the Prevention Impact and Efficiency (PIE) model provided by Pletcher *and colleagues* [19]. We also calculated individual NNT (iNNT = 1/iARR) to assess individual-level efficiency. The iNNT represented the number needed to treat (NNT) for individuals with specific characteristics over a specified period (e.g., 10 years) to prevent one CVD event [20]. The maximum iNNT within each strategy was noted, which was defined as the NNT corresponding to the individual with the highest iNNT within that strategy [19].

The distributions of CVD events averted, and the average NNT were estimated using a bootstrapping method [21]. Area-proportional Venn diagrams were used to illustrate the concordance of populations eligible for statin therapy under the two strategies with similar CVD events prevented. A Sankey diagram shows shifts between risk and benefit categories. During revision, we generated density plots to compare the distributions of predicted iARR and 10-year absolute risk between the primary strategies, calculating medians and ranges (from the minimum to the maximum) of the iARR.

To further explore treatment efficiency across different subpopulations identified by the strategies, particularly in the context of the minimum benefit threshold analyses, we conducted post hoc comparisons of average NNT across subgroups defined by concordance/discordance between absolute risk-based and individualized benefit-based eligibility criteria (e.g., high-risk only, high-benefit only, both, and the additional group eligible only under the minimum benefit threshold).

This study followed the Guidelines for Accurate and Transparent Health Estimates Reporting (GATHER) Statement (S1 GATHER Checklist) [22]. Missing values for variables used in the 2019 WHO laboratory-based equations [15] were imputed using the multiple imputations by chained equations method [23] (S1 Text). The CHARLS sampling weights were applied to all analyses to ensure national representativeness. All statistical analyses were conducted using Stata/MP version 16.0 (StataCorp LLC, College Station, TX, USA).

Based on prior evidence suggesting potential advantages in younger adults, we performed subgroup analyses restricted to participants aged 40–60 years during the revision process. Several sensitivity analyses were performed to assess the robustness of the results: (i) assuming a lower statin effect (30%) on LDL-C reduction [18]; and (ii) expanding the data source to include not only individuals with blood samples but also other individuals from the 2015 cross-sectional sample of the CHARLS. The original statistical analysis plan is presented in S2 Text. Following reviewer feedback, additional post hoc analyses were incorporated during the revision process (S3 Text). This separation ensures transparency in distinguishing pre-specified and post-review analyses.

## Results

A total of 7,287 individuals aged 40–80 years were included in the primary analysis, forming a cohort reflective of 324.6 million Chinese adults after applying population weights. The weighted population had a mean age of 57 years, with 51.7% being women (S1 Table). The demographic characteristics of individuals selected from the entire 2015 cross-sectional sample, with a mean age of 57 years and 51.4% women, were comparable to those selected from the blood sample (S2 Table).

Applying the conventional high-risk threshold (≥10% 10-year risk), the absolute risk-based approach identified 50.3 million eligible adults (15.5% of the population), with a projected 2.16 million CVD events averted over 10 years (average NNT = 23) (Table 1). To achieve a comparable number of averted events (2.19 million), the individualized benefit approach required an iARR threshold of ≥3.4% ("high-benefit"). This strategy identified a slightly smaller eligible population (49.2 million, 15.1%) with a similar average NNT (22) (Table 1). When expanding eligibility to include the intermediate-risk group (≥7.5% risk), the risk-based approach identified 78.9 million adults (24.3%) and averted 3.04 million events (NNT = 26) (S3 Table). The corresponding individualized benefit approach calibrated for similar event prevention (requiring iARR ≥ 2.8%, "at least moderate-benefit") identified 76.7 million adults (23.6%) and averted 3.05 million events (NNT = 25) (S3 Table).

**Table 1. Statin eligibilities, prevented CVD events, and efficiency of the individualized benefit approach compared with treating high-risk group.**

| | Absolute risk-based approach | Individualized benefit approach | |
| --- | --- | --- | --- |
| | Treat if high risk (score ≥ 10%) | Treat if high benefit (iARR ≥ 3.4%) | Treat if gain at least a minimum benefit as the high-risk groups (iARR ≥ 2.5%) |
| **Population-level** | | | |
| CVD events averted (in thousands) | 2161.0 (1978.9,2378.8) | 2189.0 (2013.6,2373.3) | 3566.7 (3355.2,3815.3) |
| Projected adult statin eligible (in millions) | 50.3 (46.0,54.6) | 49.2 (45.3,53.0) | 96.3 (90.7,102.0) |
| Proportion of statin eligible (%) | 15.5 (14.2,16.9) | 15.1 (14.0,16.4) | 29.7 (28.0,31.4) |
| Average NNT | 23 (23,24) | 22 (22,23) | 27 (27,27) |
| **Individual-level** | | | |
| iARR | 4.2 (2.5,9.5) | 4.3 (3.4,9.5) | 3.4 (2.5,9.5) |
| Maximum iNNT | 41 | 29 | 40 |

Point estimates and 95% CIs were reported, except the values of iARR were reported as median (the range from minimum to maximum). An iARR threshold of 3.4% would avert a similar number of CVD events to the absolute risk-based approach when treating people in the high-risk group. An iARR of 2.5% is consistent with the high-risk group. The CVD risk prediction was based on the 2019 World Health Organization laboratory-based equations incorporating age, sex, systolic blood pressure, total cholesterol, smoking status, and diabetes status [15]. Statin treatment effects were derived from the Cholesterol Treatment Trialists' Collaboration meta-analysis [34], reflecting outcomes from multiple randomized controlled trials. CVD indicates cardiovascular diseases; NNT, number needed to treat; iARR, individual absolute risk reduction; CI, confidence interval.

The superior of the average NNT under the individualized benefit approach was slightly more pronounced in younger adults aged 40–60 years: to prevent similar CVD events, the NNT of the absolute risk-based strategy treating individuals in the high-risk group was 25 (95% confidence interval [CI]: 24,27), while the individualized benefit approach treating individuals in the high-benefit group was smaller at 23 (95% CI: 22,24) (Table 2). In this age group, a similar pattern was also observed when using the intermediate-risk and moderate-benefit cut-offs: the average NNT was 30 (95% CI: 29,31) if treating those at least intermediate risk (score ≥ 7.5%), and 28 (95% CI: 27,29) when treating those at least moderate benefit (S4 Table).

Applying the Causal-Benefit model's minimum benefit threshold principle, using the minimum iARR observed in the high-risk group (iARR ≥ 2.5% in our population), the individualized benefit approach substantially expanded eligibility compared to the standard high-risk approach (29.7% [96.3 million] versus 15.5% [50.3 million]) and consequently prevented more events (3.57 million versus 2.16 million) (Table 1). Analysis of NNT across subgroups defined by risk/benefit status revealed varying efficiency levels (Fig 2): NNT was lowest for individuals meeting both high-risk and high-benefit criteria (NNT = 22), intermediate for those meeting only high-benefit (NNT = 26) or only high-risk (NNT = 33) criteria, and highest for the lower-risk/lower-benefit group added by applying the minimum iARR threshold (NNT = 35). This suggests potential efficiency gradients for prioritization.

Discordance between the strategies (when calibrated for similar event prevention) was substantial. Among the 58.9 million adults recommended for statin therapy by either strategy, comparing high-risk versus high-benefit groups, the overlap was only 68.9% (Fig 3). Given that the two approaches prevent a similar number of events, beyond the overlap between the two approaches, the risk-based strategy uniquely identified 9.7 million (16.5%) individuals (high-risk/lower-benefit), while the benefit strategy uniquely identified 8.6 million (14.6%) individuals (lower-risk/high-benefit). For the at least intermediate-risk versus at least moderate-benefit comparison, overlap increased to 76.5%, but the expanded risk-based approach still missed 9.3 million selected by the benefit approach (Fig 3). A Sankey diagram (Fig 4) illustrates transitions between the absolute risk categories and risk reduction groups. Notably, 1.3 million individuals were reclassified from the

**Table 2. Statin eligibilities, prevented CVD events, and efficiency of the individualized benefit approach compared with treating high-risk group (adults aged 40–60 years).**

| | Absolute risk-based approach | Individualized benefit approach | |
|---|---|---|---|
| | Treat if high risk (score ≥ 10%) | Treat if high benefit (iARR ≥ 3.8%) | Treat if gain at least a minimum benefit as the high-risk groups (iARR ≥ 2.5%) |
| **Population-level** | | | |
| CVD events averted (in thousands) | 88.1 (63.2,114.3) | 96.6 (65.5,133.0) | 574.2 (487.0,686.5) |
| Projected adult statin eligible (in millions) | 2.2 (1.6,2.8) | 2.2 (1.5,2.9) | 18.5 (15.1,21.8) |
| Proportion of statin eligible (%) | 1.1 (0.8,1.4) | 1.1 (0.8,1.5) | 8.8 (7.3,10.6) |
| Average NNT | 25 (24,27) | 23 (22,24) | 32 (31,33) |
| **Individual-level** | | | |
| iARR | 4.0 (2.5,6.5) | 4.3 (3.8,6.5) | 2.9 (2.5,6.5) |
| Maximum iNNT | 41 | 26 | 40 |

Point estimates and 95% CIs were reported. An iARR threshold of 3.8% would avert a similar number of CVD events to the absolute risk-based approach when treating people in the high-risk group. An iARR of 2.5% is consistent with the minimum iARR of the high-risk group. The CVD risk prediction was based on the 2019 World Health Organization laboratory-based equations incorporating age, sex, systolic blood pressure, total cholesterol, smoking status, and diabetes status [15]. Statin treatment effects were derived from the Cholesterol Treatment Trialists' Collaboration meta-analysis [34], reflecting outcomes from multiple randomized controlled trials. CVD indicates cardiovascular diseases; NNT, number needed to treat; iARR, individual absolute risk reduction; CI, confidence interval.

borderline risk group (10-year CVD risk 5% to 7.5%) to the high-benefit group (iARR ≥ 3.4%), while 2.1 million were reclassified from the high-risk group (10-year CVD risk ≥10%) to the low-benefit group (iARR < 2.8%).

Comparisons of the iARR distributions revealed differences in targeting efficiency. Although the median iARR was similar between the high-risk and high-benefit groups (Table 1), the distribution for the iARR in the high-risk group exhibited a longer lower end-tail compared to the high-benefit group: the lower bound was 2.5% versus 3.4% for the absolute risk-based and the individualized benefit approach respectively while the upper bound were both at 9.5%. The corresponding maximum iNNT values were 41 for the risk-based approach versus 29 for the individualized benefit approach. (S2B Fig and Table 1). This indicates the risk-based approach selects a larger proportion of individuals with low predicted benefit. Critically, 19.3% (9.7 million in 50.3 million) selected by the risk-based approach had iARR < 3.4% (iNNT > 29), whereas none selected by the benefit approach fell below this threshold (S2B Fig). The difference in iARR distribution tails was particularly evident in younger adults (aged 40–60 years): the lower bound of the iARR under the risk-based approach and the individualized benefit approach was 2.5% and 3.8%, respectively, and the maximum iNNT values were 41 and 26, correspondingly (S2D Fig). This demonstrates the potential superior efficiency of the individualized benefit approach in excluding individuals predicted to derive minimal absolute benefit from statin therapy.

Consistent with differential targeting, individuals uniquely identified by the high-benefit approach (lower-risk/high-benefit) were significantly younger, and had higher mean TC and LDL-C levels compared to those uniquely identified by the high-risk approach (high-risk/lower-benefit) (P < 0.001 for all comparisons, S5 Table). In the younger adult subgroup specifically, the benefit-only group also had lower mean systolic blood pressure than the risk-only group (P < 0.001, Table 3), further supporting that the benefit approach better targets individuals with higher lipid-driven risk.

Fig 5 provides illustrative examples relevant to shared decision-making. Under the absolute risk-based approach, individuals A and B, both exceeding the 10% high-risk threshold, would typically be recommended statins, while individual C (risk <10%) would not. However, the individualized benefit approach, considering predicted iARR, would recommend treatment for individuals A (iARR 4.90%) and C (iARR 3.67% > 3.4% threshold) but not for individual B (iARR 2.80% < 3.4% threshold), despite B having the same high absolute risk as A. Fig 6 illustrates how the individualized benefit approach prioritizes individuals based on treatment benefits rather than absolute risk alone.

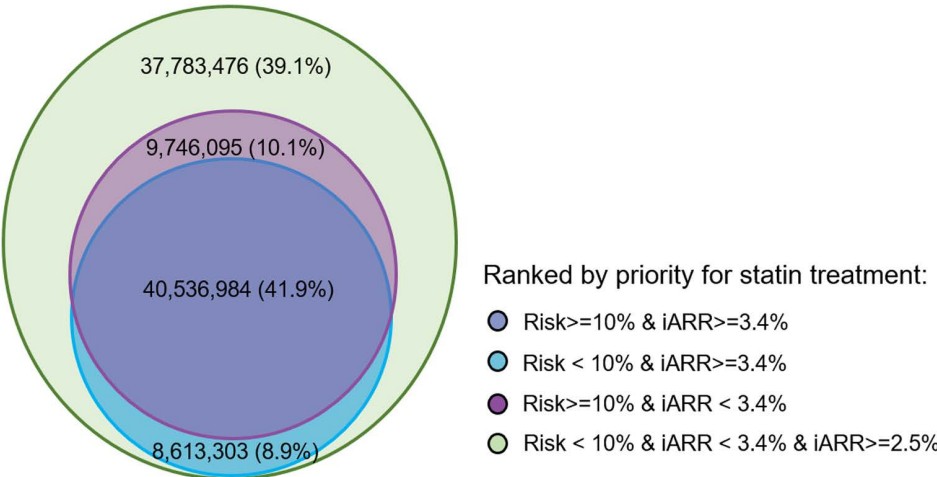

| Priority | Statin treatment subgroups | CVD events averted | Average NNT |
|---|---|---|---|
| 1 | Risk>=10% & iARR>=3.4% | 1,863,164 | 22 |
| 2 | Risk < 10% & iARR>=3.4% | 325,820 | 26 |
| 3 | Risk>=10% & iARR < 3.4% | 297,797 | 33 |
| 4 | Risk < 10% & iARR < 3.4% & iARR>=2.5% | 1,088,841 | 35 |

**Fig 2. The treatment efficiency of subgroups with different benefits and risk levels.** Numbers outside the parentheses indicate the number of statin-treated individuals in each color-coded subgroup; numbers inside the parentheses indicate their proportion relative to the total statin-treated population of gaining at least a minimum benefit as the high-risk groups (iARR ≥ 2.5%). iARR indicates individual absolute risk reduction; CVD, cardiovascular diseases; NNT, number needed to treat.

In sensitivity analyses, assuming a 30% LDL-C reduction, the "high-benefit" threshold for the individualized benefit approach decreased from 3.4% to 2.7%, with the proportion of overlap between the two strategies reduced from 68.9% to 66.3% (S3 Fig, S6 and S7 Tables). Results were consistent when the data was expanded to total participants from the 2015 cross-sectional sample, indicating the robustness of the findings (S3 Fig, S8 and S9 Tables).

## Discussion

To the best of our knowledge, this is the first study using nationally representative data from China to estimate statin eligibility and efficiency across different strategies for primary prevention. Our study suggested that while the absolute risk-based and individualized benefit approaches can be calibrated to achieve similar population-level outcomes, they select fundamentally different groups of individuals for treatment. We observed a surprisingly low degree of overlap, with a large proportion of individuals being uniquely identified by only one of the two strategies. Although broadly similar population-level performance was achieved, these two approaches identified two groups of target individuals, with only two-thirds of them overlapping. The individualized benefit approach shows considerably better performance in individual-level evaluations, such as iNNT, which demonstrates the number needed to treat for individuals with different characteristics. This approach can efficiently support shared decision-making for statin recommendations, particularly in

**(A) Total population**

Treat if high risk (score ≥ 10%) *vs.*
Treat if high benefit (iARR ≥ 3.4%)

Treat if intermediate risk and higher (score ≥ 7.5%) *vs.*
Treat if moderate benefit and higher (iARR ≥ 2.8%)

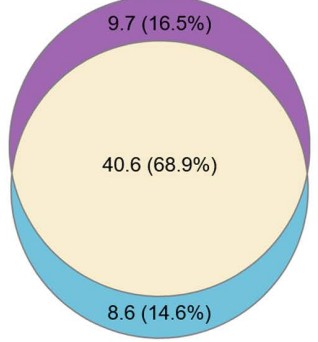

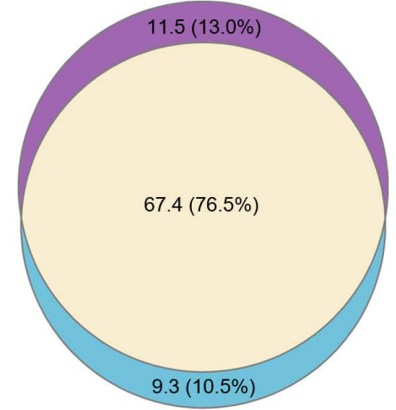

**(B) Adults aged 40-60**

Treat if high risk (score ≥ 10%) *vs.*
Treat if high benefit (iARR ≥ 3.8%)

Treat if intermediate risk and higher (score ≥ 7.5%) *vs.*
Treat if moderate benefit and higher (iARR ≥ 3.0%)

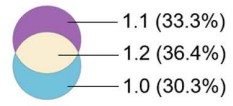

1.1 (33.3%)
1.2 (36.4%)
1.0 (30.3%)

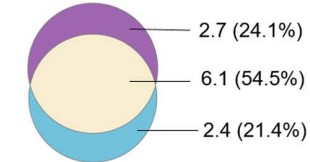

2.7 (24.1%)
6.1 (54.5%)
2.4 (21.4%)

● Absolute risk-based approach only
○ Individualized benefit approach only

**Fig 3. Discordance in statin eligibility by different approaches with comparable CVD events averted.** The absolute number of individuals eligible for statin therapy (in millions) and the corresponding percentages (%) were reported. **(A)** A threshold of iARR of 2.8% would avert a similar number of CVD events to the absolute risk-based approach when treating people in intermediate-risk and high-risk groups; a threshold of iARR of 3.4% would avert a similar number of CVD events to the absolute risk-based approach when treating people in the high-risk group. **(B)** For younger persons ages 40–60 years, a threshold of iARR of 3.0% would avert a similar number of CVD events to the absolute risk-based approach when treating people in intermediate-risk and high-risk groups; a threshold of iARR of 3.8% would avert a similar number of CVD events to the absolute risk-based approach when treating people in high-risk group. iARR indicates individual absolute risk reduction; CVD, cardiovascular diseases.

LMICs. Moreover, an alternative strategy, which combines the absolute risk-based and individualized benefit approaches with lower thresholds, can gain substantial benefits by identifying more individuals eligible for statin.

Our results are consistent with growing international evidence from studies in the Scottish [24] and the United States (U.S.) [19] populations, demonstrating that the individualized benefit approach is more efficient than the absolute risk-based approach in CVD primary prevention. The difference in the number of statins treated under the two strategies in the Chinese population was comparable to the Scottish population and appeared more pronounced than the U.S. population. We standardized intervention populations across studies in the U.S., Scotland, and China to prevent 100,000 CVD events. The difference in statin treatments guided by absolute risk-based versus individualized benefit strategies was 50,000 in the U.S. [19], 99,051 in Scotland [24], and 81,538 in China. Critically, the iARR distribution confirms better individual-level targeting efficiency for the benefit approach in our population. Given the demographic context of LMICs, the population-level implications of adopting benefit-based approaches are substantial.

Using a 10% cut-off for the 10-year CVD risk score under the absolute risk-based approach and a 3.4% ARR threshold under the individualized benefit approach, we found that 17.5% (8.6 million in 49.2 million) of individuals with an expected ARR ≥ 3.4% (equivalent to an individual NNT < 30) would be undertreated if guided by absolute risk. This discordance

PLOS Medicine

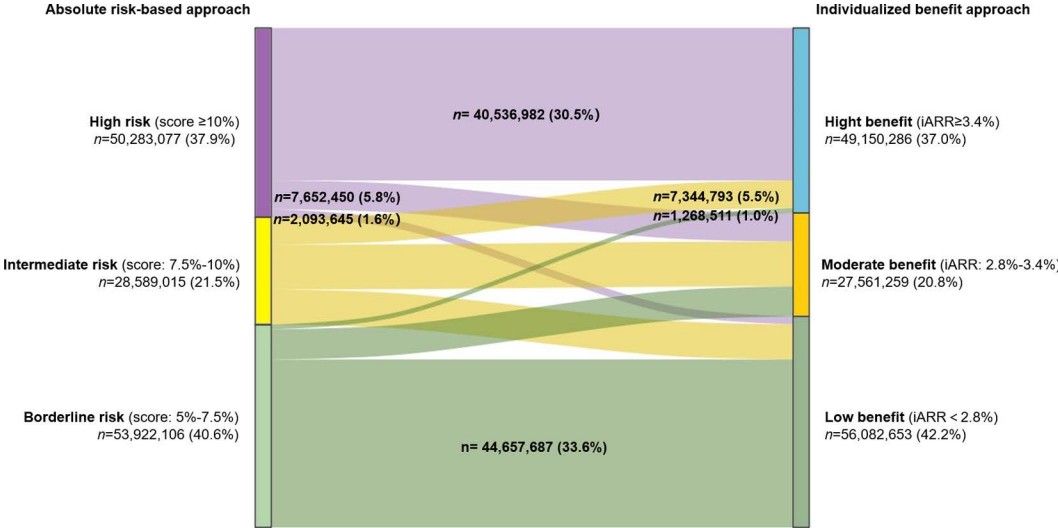

**Fig 4. Switchover from absolute risk-based to individualized benefit approach.** Under the individualized benefit approach, a threshold of iARR of 2.8% would avert a similar number of CVD events to the absolute risk-based approach when treating people in intermediate-risk and high-risk groups; a threshold of iARR of 3.4% would avert a similar number of CVD events to the absolute risk-based approach when treating people in high-risk group. CVD, cardiovascular disease; iARR, individual absolute risk reduction.

**Table 3. Comparisons of the baseline characteristics of the discordance groups by different strategies (aged 40–60 years).**

| Characteristics | Risk reduction ≥3.8% and absolute risk <10% | Risk reduction <3.8% and absolute risk ≥10% | *P*-value |
|---|---|---|---|
| Age (years) | 56.3 (3.9) | 56.2 (4.0) | 0.972 |
| Men | 94.5 | 100.0 | 0.205 |
| Current smoking | 88.4 | 97.8 | 0.130 |
| Hypertension | 81.2 | 100.0 | 0.078 |
| SBP (mmHg) | 144.6 (16.4) | 165.0 (14.6) | <0.001 |
| DBP (mmHg) | 84.8 (8.1) | 97.2 (11.4) | <0.001 |
| TC (mmol/L) | 6.2 (0.4) | 4.8 (0.8) | <0.001 |
| LDL-C (mmol/L) | 4.0 (0.4) | 2.4 (0.4) | <0.001 |
| HDL-C (mmol/L) | 1.4 (0.2) | 1.6 (0.7) | 0.530 |

Values are mean (SD) or %. SBP indicates systolic blood pressure; DBP, diastolic blood pressure; TC, total cholesterol; LDL-C, low-density lipoprotein cholesterol; HDL-C, high-density lipoprotein cholesterol.

highlights that higher absolute risk does not necessarily align with greater treatment benefits from statin therapy. Clinical guidelines, such as the 2021 European Society of Cardiology (ESC) guideline [3], emphasize the importance of informed discussions tailored to individual needs. From a shared decision-making perspective, prioritizing individuals likely to derive the greatest benefit aligns with guideline emphases on individualized care. Our findings suggest incorporating predicted benefit (iARR or iNNT) into discussions about statin initiation may be more informative than relying solely on absolute risk.

In LMICs, where statin use is low despite a large number of eligible individuals, efficiently identifying suitable individuals is critical. Transitioning to an individualized benefit approach substantially alters the eligible population profile, potentially aligning better with priorities in resource-limited settings. Integrating predicted benefits into shared decision-making, supported by the Causal-Benefit model and other tools, offers a tailored approach [11]. Applying the Causal-Benefit model's minimum benefit threshold (iARR ≥ 2.5%) substantially increased events prevented (65%) by expanding eligibility,

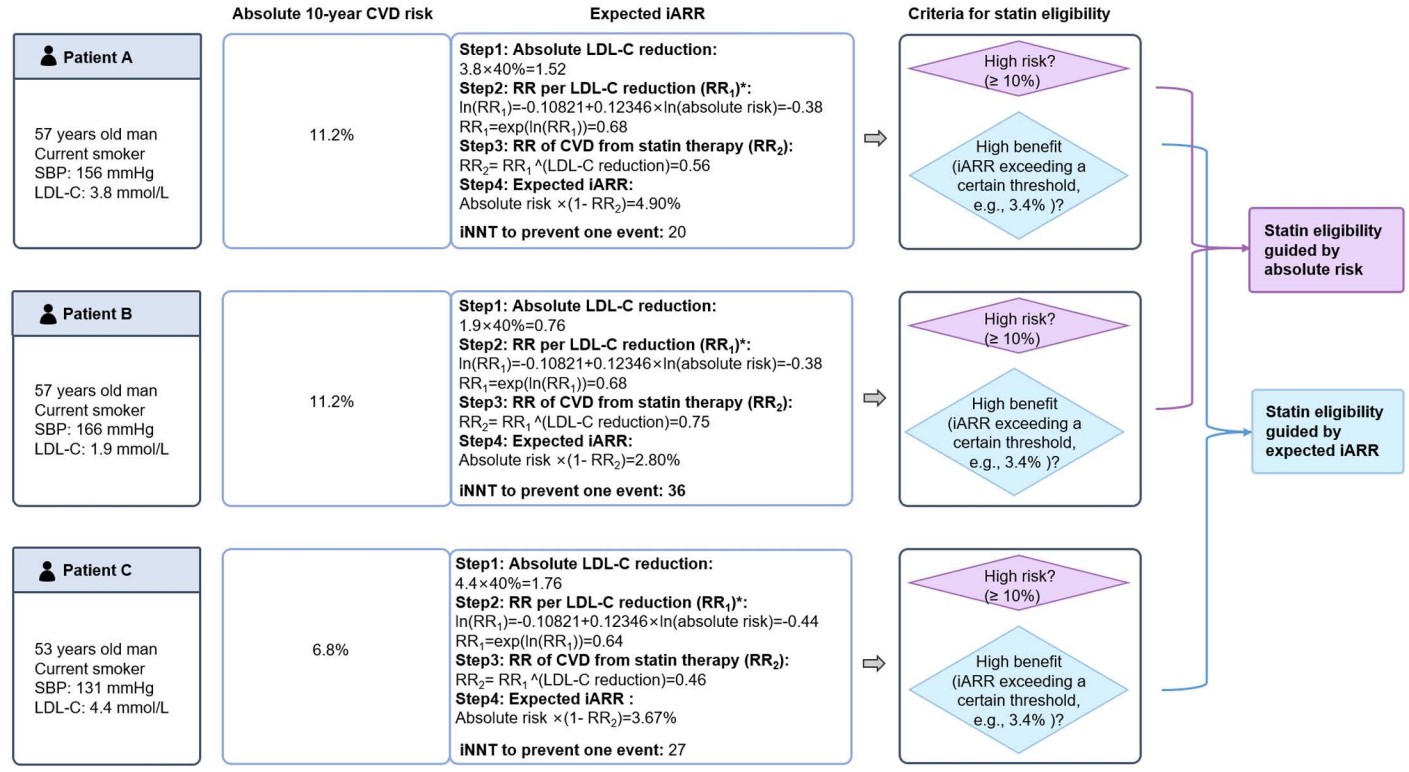

**Fig 5. Patient example to illustrate statin eligibility guided by the absolute risk and the individualized benefit.** *To assess the RR (relative risk) of CVD per unit of LDL-C reduction, we used methods proposed by Thanassoulis and colleagues [10]. CVD, indicates cardiovascular disease; iARR, individual absolute risk reduction; LDL-C, low-density lipoprotein cholesterol; SBP, systolic blood pressure; NNT, number needed to treat; iNNT, individual NNT.

demonstrating the model's potential for enhancing overall prevention. However, our subgroup analysis revealed varying efficiency levels within this expanded group. Therefore, considering budget constraints in LMICs, healthcare context-specific iARR thresholds are crucial to balance maximal prevention benefits with efficient resource allocation.

Predicted risk reduction (iARR) integrates with risk prediction models and can be expressed as iNNT, potentially offering an intuitive metric for treatment benefit discussions [20]. Our findings that the benefit approach uniquely identifies younger individuals, more women, and those with higher LDL-C align with prior studies in the U.S. [10] and Brazilian [25] populations. This supports prioritizing earlier intervention based on benefit, considering cumulative LDL-C exposure and greater long-term benefit. Early intervention in younger adults is critical, as cumulative LDL-C exposure during young and middle adulthood is associated with increased CVD risk [26], and the benefits of LDL-C lowering are more substantial with prolonged treatment [27]. Individualized benefit approaches incorporating long-term [28] or lifetime perspectives [29] may be particularly useful for guiding decisions in younger adults compared to standard 10-year models.

Previous studies have also noted that high-risk individuals tend to be older men with elevated systolic blood pressure and high smoking rates [30]. Women, who often have higher cholesterol levels but lower short-term absolute risk, are less frequently recommended for statins under current guidelines [31]. By prioritizing individuals based on treatment benefits rather than absolute risk, the individualized benefit approach has the potential to expand statin use among younger adults and women, addressing critical gaps in current practice.

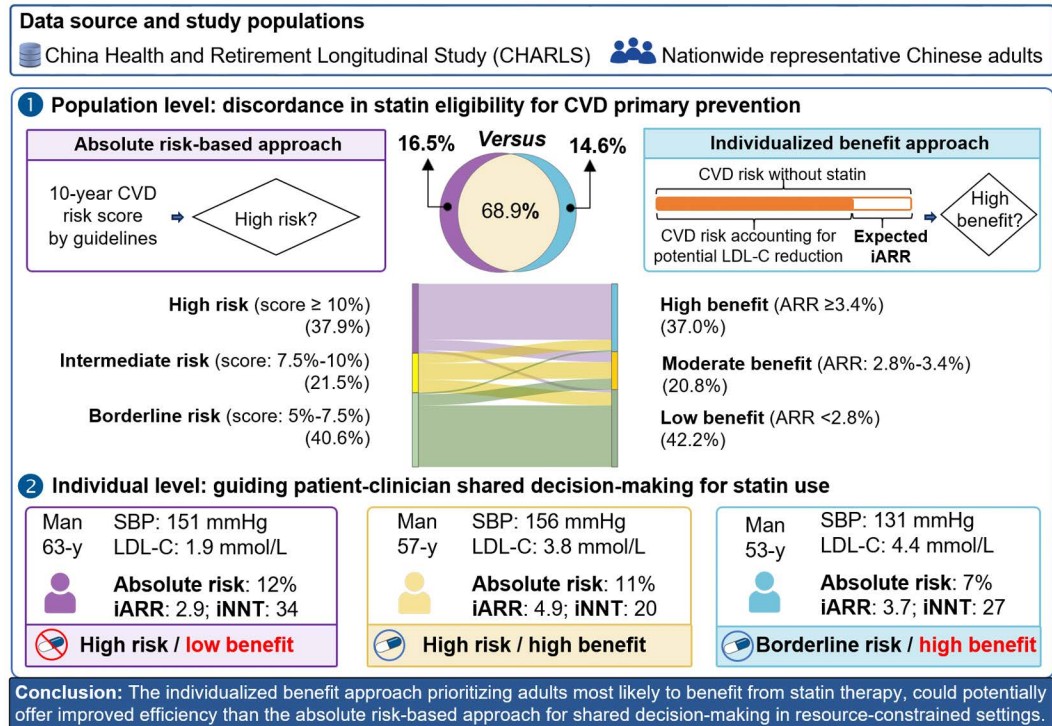

**Fig 6. Illustration of how the individualized benefit approach prioritizes individuals based on treatment benefits rather than absolute risk alone.** Absolute risk-based approach: statin eligibility based on 10-year CVD risk assessment by current guidelines. Individualized benefit approach: accounting for the potential benefits of lipid-lowering therapy based on expected iARR. CVD, indicates cardiovascular disease; iARR, individual absolute risk reduction; LDL-C, low-density lipoprotein cholesterol; SBP, systolic blood pressure; iNNT, individual number needed to treat.

The major strength of this study is the use of nationally representative data from China, enabling us to assess the numbers of statin eligibilities, the CVD events prevented, and the number needed to treat for various statin prevention strategies in a broad population. However, there are several limitations to address. First, our estimates of CVD events averted and NNTs relied on assumptions, including the use of WHO laboratory-based risk equations, which may not fully capture future CVD risk in China. We recalibrated the models to adjust for risk overestimation based on previous evaluations [16]. Second, consistent with prior modeling studies [10,19,32,33], we assumed quantifiable benefit stems primarily from LDL-C reduction. While LDL-C lowering is the principal mechanism for statin benefit [34,35], this simplification overlooks other effects, such as plaque stabilization [36], and may underestimate benefits, particularly for high-risk individuals with lower LDL-C levels. However, our modeling approach is anchored in the well-established finding from the Cholesterol Treatment Trialists' (CTT) Collaboration that the reduction in cardiovascular event rates is approximately constant per mmol/L reduction in LDL-C [34]. This mathematical relationship enables robust estimation of expected benefits, irrespective of the underlying biological mechanisms. An alternative strategy, aligned with the principles of the Causal-Benefit model [11], is to apply a minimum benefit threshold to expand eligibility rather than only optimizing the selection of a fixed-size group. As demonstrated in our results, this strategy could substantially increase the number of cardiovascular events prevented by identifying a much larger number of eligible individuals. Other limitations include assuming constant treatment effects over time. We modeled only 10-year benefits; longer-term models might show greater efficiency, especially for younger adults [28]. We modeled only moderate-intensity statins; individual benefits vary with statin type and intensity.

Finally, the potential for improved targeting efficiency at both population and individual levels warrants further research on cost-effectiveness.

In summary, the individualized benefit approach prioritizes individuals most likely to benefit from statin therapy, enhancing the capacity to discriminate treatment effects at the individual level, particularly in populations with distinct cardiovascular risk profiles. Adopting benefit-based principles to guide statin eligibility in LMICs, where large populations stand to benefit, supports shared decision-making for statin recommendations, fostering informed, patient-centered clinical decisions.

## Supporting information

**S1 Text. Supplementary methods.**
(DOCX)

**S2 Text. Statistical analysis plan.**
(DOCX)

**S3 Text. Post-peer review analytical enhancements.**
(DOCX)

**S1 GATHER Checklist.** Reporting checklist of items for heath estimates study based on the GATHER guideline.
(DOCX)

**S1 Fig. Flow chart of inclusion of study participants.** CHARLS indicates the China Health and Retirement Longitudinal Survey; CVD, cardiovascular disease, LDL-C, low-density lipoprotein cholesterol.
(TIF)

**S2 Fig. Distribution of the iARR under the absolute risk-based approach and the individualized benefit approach.**
**(A)** In the total population, an iARR threshold of 3.4% in the total population, would avert a similar number of cardiovascular events to the absolute risk-based strategy when treating people in the high-risk group. **(B)** In the total population, an iARR threshold of 2.8% would avert a similar number of cardiovascular events to the absolute risk-based strategy approach when treating people in the intermediate- and high-risk groups. **(C)** For younger adults aged 40–60 years, an iARR threshold of 3.8% would avert a similar number of cardiovascular events to the absolute risk-based strategy when treating people in the high-risk group. **(D)** In adults aged 40–80 years, an iARR threshold of 3.0% would avert a similar number of cardiovascular events to the absolute risk-based strategy when treating people in intermediate- and high-risk groups. iARR indicates individual absolute risk reduction.
(TIF)

**S3 Fig. Estimation of the overlap in statin eligibility across different approaches with similar CVD events prevented (sensitivity analysis).** The absolute number of individuals eligible for statin therapy (in millions) and the corresponding percentages (%) were reported. **(A)** Under the individualized benefit approach, a 2.7% threshold would avert a similar number of CVD events as the absolute risk-based approach when treating people in high-risk group. **(B)** A 3.4% threshold of the individualized benefit approach would avert a similar number of CVD events as the absolute risk-based approach when treating people in high-risk group. CVD indicates cardiovascular disease, LDL-C, low-density lipoprotein cholesterol.
(TIF)

**S1 Table. Weighted characteristics of participants aged 40–80 years (main analysis).** Values are mean (SD) or % unless otherwise noted. ªPresented as median (IQR). SBP indicates systolic blood pressure; DBP, diastolic blood

pressure; TC, total cholesterol; LDL-C, low-density lipoprotein cholesterol; HDL-C, high-density lipoprotein cholesterol; IQR, interquartile range.
(DOCX)

**S2 Table. Weighted characteristics of participants aged 40–80 years (expanding data source to the entire 2015 cross-sectional sample).** Values are mean (SD) or % unless otherwise noted. [a]Presented as median (IQR). SBP indicates systolic blood pressure; DBP, diastolic blood pressure; TC, total cholesterol; LDL-C, low-density lipoprotein cholesterol; HDL-C, high-density lipoprotein cholesterol; IQR, interquartile range.
(DOCX)

**S3 Table. Statin eligibilities, prevented CVD events, and efficiency of the individualized benefit approach compared with treating intermediate- and high-risk groups.** Point estimates and 95% CIs were reported, except the values of iARR were reported as median (the range from minimum to maximum). An iARR threshold of 2.8% would avert a similar number of CVD events to the absolute risk-based approach when treating people in the intermediate- and high-risk groups. An iARR of 2.0% is consistent with the minimum iARR of the intermediate- and high-risk groups. The CVD risk prediction was based on the 2019 World Health Organization laboratory-based equations incorporating age, sex, systolic blood pressure, total cholesterol, smoking status, and diabetes status [15]. Statin treatment effects were derived from the Cholesterol Treatment Trialists' Collaboration meta-analysis [34], reflecting outcomes from multiple randomized controlled trials. CVD indicates cardiovascular diseases; NNT, number needed to treat; iARR, individual absolute risk reduction; CI, confidence interval.
(DOCX)

**S4 Table. Statin eligibilities, prevented CVD events, and efficiency of the individualized benefit approach compared with treating intermediate- and high-risk groups (aged 40–60 years).** Point estimates and 95% CIs were reported, except the values of iARR were reported as median (the range from minimum to maximum). An iARR threshold of 3.0% would avert a similar number of CVD events to the absolute risk-based strategy when treating people in the intermediate- and high-risk groups. An iARR of 2.0% is consistent with the minimum iARR of the intermediate- and high-risk groups. The CVD risk prediction was based on the 2019 World Health Organization laboratory-based equations incorporating age, sex, systolic blood pressure, total cholesterol, smoking status, and diabetes status [15]. Statin treatment effects were derived from the Cholesterol Treatment Trialists' Collaboration meta-analysis [34], reflecting outcomes from multiple randomized controlled trials. CVD indicates cardiovascular diseases; NNT, number needed to treat; iARR, individual absolute risk reduction; CI, confidence interval.
(DOCX)

**S5 Table. Comparisons of the baseline characteristics of the discordance groups by different strategies.** Values are mean (SD) or %. SBP indicates systolic blood pressure; DBP, diastolic blood pressure; TC, total cholesterol; LDL-C, low-density lipoprotein cholesterol; HDL-C, high-density lipoprotein cholesterol.
(DOCX)

**S6 Table. Statin eligibilities, prevented CVD events, and efficiency of the individualized benefit approach compared with treating high-risk group (assuming a lower statin effect of 30% on LDL-C reduction).** Point estimates and 95% CIs were reported, except the values of iARR were reported as median (the range from minimum to maximum). An iARR threshold of 2.7% would avert a similar number of CVD events to the absolute risk-based strategy when treating people in the high-risk group. An iARR of 2.0% is consistent with the minimum iARR of the high-risk group. The CVD risk prediction was based on the 2019 World Health Organization laboratory-based equations incorporating age, sex, systolic blood pressure, total cholesterol, smoking status, and diabetes status [15]. Statin treatment effects were derived from the Cholesterol Treatment Trialists' Collaboration meta-analysis [34], reflecting outcomes from multiple randomized controlled

trials. CVD indicates cardiovascular diseases; NNT, number needed to treat; iARR, individual absolute risk reduction; CI, confidence interval.
(DOCX)

**S7 Table. Statin eligibilities, prevented CVD events, and efficiency of the individualized benefit approach compared with treating intermediate- and high-risk groups (assuming a lower statin effect of 30% on LDL-C reduction).** Point estimates and 95% CIs were reported, except the values of iARR were reported as median (the range from minimum to maximum). An iARR threshold of 2.2% would avert a similar number of CVD events to the absolute risk-based strategy when treating people in the intermediate- and high-risk groups. An iARR of 1.6% is consistent with the minimum iARR of the intermediate- and high-risk groups. The CVD risk prediction was based on the 2019 World Health Organization laboratory-based equations incorporating age, sex, systolic blood pressure, total cholesterol, smoking status, and diabetes status [15]. Statin treatment effects were derived from the Cholesterol Treatment Trialists' Collaboration meta-analysis [34], reflecting outcomes from multiple randomized controlled trials. CVD indicates cardiovascular diseases; NNT, number needed to treat; iARR, individual absolute risk reduction; CI, confidence interval.
(DOCX)

**S8 Table. Statin eligibilities, prevented CVD events, and efficiency of the individualized benefit approach compared with treating high-risk group (expanding data source to the entire 2015 cross-sectional sample).** Point estimates and 95% CIs were reported, except the values of iARR were reported as median (the range from minimum to maximum). An iARR threshold of 3.4% would avert a similar number of CVD events to the absolute risk-based strategy when treating people in the high-risk group. An iARR of 2.5% is consistent with the minimum iARR of the high-risk group. The CVD risk prediction was based on the 2019 World Health Organization laboratory-based equations incorporating age, sex, systolic blood pressure, total cholesterol, smoking status, and diabetes status [15]. Statin treatment effects were derived from the Cholesterol Treatment Trialists' Collaboration meta-analysis [34], reflecting outcomes from multiple randomized controlled trials. CVD indicates cardiovascular diseases; NNT, number needed to treat; iARR, individual absolute risk reduction; CI, confidence interval.
(DOCX)

**S9 Table. Statin eligibilities, prevented CVD events, and efficiency of the individualized benefit approach compared with treating intermediate and high-risk groups (expanding data source to the entire 2015 cross-sectional sample).** Point estimates and 95% CIs were reported, except the values of iARR were reported as median (the range from minimum to maximum). An iARR threshold of 2.8% would avert a similar number of CVD events to the absolute risk-based strategy when treating people in the intermediate- and high-risk groups. An iARR of 2.0% is consistent with the minimum iARR of the intermediate- and high-risk groups. The CVD risk prediction was based on the 2019 World Health Organization laboratory-based equations incorporating age, sex, systolic blood pressure, total cholesterol, smoking status, and diabetes status [15]. Statin treatment effects were derived from the Cholesterol Treatment Trialists' Collaboration meta-analysis [34], reflecting outcomes from multiple randomized controlled trials. CVD indicates cardiovascular diseases; NNT, number needed to treat; iARR, individual absolute risk reduction; CI, confidence interval.
(DOCX)

## Acknowledgments

The authors thank the China Health and Retirement Longitudinal Study (CHARLS) team for the public data they provided, and we appreciate all the staff, investigators and the participation of respondents in the CHARLS study. We also thank the authors for providing the proposed open-source prevention impact and efficiency (PIE) model (https://github.com/markpletcher/PIE-Model_Stata).

## Author contributions

**Conceptualization:** Qiuping Liu, Xun Tang, Pei Gao.

**Data curation:** Qiuping Liu, Chao Gong.

**Formal analysis:** Qiuping Liu.

**Funding acquisition:** Xun Tang, Pei Gao.

**Methodology:** Qiuping Liu, Xiaofei Liu, Xun Tang, Pei Gao.

**Project administration:** Xun Tang, Pei Gao.

**Resources:** Xun Tang, Pei Gao.

**Software:** Qiuping Liu.

**Supervision:** Xun Tang, Pei Gao.

**Validation:** Qiuping Liu, Chao Gong, Tianjing Zhou, Minglu Zhang.

**Visualization:** Qiuping Liu.

**Writing – original draft:** Qiuping Liu.

**Writing – review & editing:** Qiuping Liu, Chao Gong, Tianjing Zhou, Minglu Zhang, Xiaofei Liu, Xun Tang, Pei Gao.

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
