## [Editor Report · Decision Letter 0]

Dear Dr Gao, 

Thank you for submitting your manuscript entitled "Absolute risk-based versus risk reduction-based strategies for determining statin eligibility in primary prevention of cardiovascular diseases in Chinese populations" for consideration by PLOS Medicine.

Your manuscript has now been evaluated by the PLOS Medicine editorial staff and I am writing to let you know that we would like to send your submission out for external peer review.

Please re-submit your manuscript within two working days, i.e. Feb 10 2025.

Feel free to email me at atosun@plos.org or us at plosmedicine@plos.org if you have any queries relating to your submission.

Kind regards,

Alexandra Tosun, PhD

Associate Editor

PLOS Medicine

---

## [Decision Letter · Decision Letter 1]

Dear Dr Gao,

Many thanks for submitting your manuscript "Absolute risk-based versus risk reduction-based strategies for determining statin eligibility in primary prevention of cardiovascular diseases in Chinese populations" (PMEDICINE-D-25-00419R1) to PLOS Medicine. The paper has been reviewed by subject experts and a statistician; their comments are included below and can also be accessed here: [LINK]

As you will see, the reviewers, while supporting a revision of the manuscript, raise serious concerns about the manuscript. After discussing the paper with the editorial team and an academic editor with relevant expertise, I'm pleased to invite you to revise the paper in response to the reviewers' comments. We plan to send the revised paper to some or all of the original reviewers, and we cannot provide any guarantees at this stage regarding publication.

We ask that you submit your revision by Apr 18 2025. However, if this deadline is not feasible, please contact me by email, and we can discuss a suitable alternative.

Don't hesitate to contact me directly with any questions (atosun@plos.org). 

Best regards, 

Alexandra 

Alexandra Tosun, PhD 

Associate Editor

PLOS Medicine

atosun@plos.org

Comments from the reviewers: 

Reviewer #1: This study re-visit the question on whether use of statin should be based on absolute risk of CVD or estimated absolute risk reduction. This is an interesting question given today's emphasis on precision prevention. The data and approach to the question appear to be robust overall. However, I have several comments: 

- The estimated number of CVD avoidable between two approaches were extremely close, with massive overlaps in 95% CIs (Table 1). The projected people eligible for statin and the average NNT were also very similar. The only discernible difference between the two methods are the maximum iNNT. However that statistic is only one person and I do wonder whether that warrants the study's strong conclusion 'offering enhanced efficiency than the absolute risk-based approach'. Also I wondered if iNNT makes sense - NNT is a measure for population, what does it tell to an individual? 

- perhaps a clearer approach to show the ARR approach to be better (for individuals) would be to examine the distribution of ARR. From the data shown (similar average NNT but different max iNNT), i'd expect the ARR would have a longer lower end tail in the absolute risk approach. 

- Critically, this study assumes the effect of statin on CVD prevention is only via LDL-c reduction. I believe this assumption requires strong justification. This is a strong assumption that could overestimate the efficiency of ARR reduction approach, since the ARR is based on LDL-c level. There are studies that showed statin not only reduces circulatory LDL-c but also stabilises plagues (https://pubmed.ncbi.nlm.nih.gov/30642643/). Instead, I wondered why the effect of statin is not modelled based on clinical trials? There are trials that estimated CVD risk reduction directly. 

Reviewer #2: This analysis compares two approaches to select individuals to prevent ASCVD. One is the conventional model- the absolute risk approach; the second is the absolute risk reduction approach, which is based on the Causal-Benefit model to prevent ASCVD.1 The analysis was designed to compare the two approaches to prevent an equal number of events. Their results show the absolute risk approach was slightly superior. Unfortunately, their analysis does not reproduce the intent of the new model- the Causal Benefit model of cardiovascular prevention because it does not reproduce the methods of the new model. Indeed, the methods of their analysis frustrate the purpose for which it was developed. I will explain.

All the accepted models to select individuals for pharmacological treatment to prevent cardiovascular disease use a risk period of 10 years. In this regard, all follow the pattern of the original model- the Framingham model risk score. I presume Framingham picked a 10-year period to predict risk because this was, at the time, the longest period for which they had reliable data. But this singular decision- to limit the prediction of ASCVD risk to 10 years- has had multiple adverse consequences, which many still fail to appreciate.

Chief among these is that the non-modifiable factors that increase cardiovascular risk- age and sex- account for the overwhelming majority of risk whereas the modifiable causal factors- the apoB lipoproteins, blood pressure, diabetes- account for little of the risk.2 This means that age and sex become the determining factors for selection. Thus, for the American Guidelines, virtually all men over 60 are eligible, whereas a much smaller number of men and almost no women under 60 are eligible.3 This is an inappropriate result for multiple reasons, not least the biases against women and those who are younger. 

More fundamentally, risk is not the same as incidence. Notwithstanding that the 10-year risk threshold is not commonly crossed until after age 60, about 40-45% of all ASCVD events occur before age 60.4 Why? If risk is so much lower in those under 60, how can the incidence of events be so close? The answer is simple. Risk is the number of events per standard number of individuals per unit time. Incidence is the total number of events. And there many more under 60 than over 60. Accordingly, larger numbers at lower risk produce large numbers with events. Furthermore, by any measure that matters, a premature heart attack or a premature stroke or a premature death or incapacity due to vascular disease is more tragic, more costly to the individual, their family and society than the same events in those who are older, who have had more opportunity to create and contribute. But there is yet another fatal flaw in the conventional model: many of the ASCVD events that occur after 60 are due to the progression of disease that developed before age 60 and continued to progress until after age 60, disease whose progression could have been prevented by proven medical therapy.

The risk model did make sense. Treat those most likely to have an event. But it is the time frame- the period of 10 years that converts the risk model from a strategy based on common sense to a strategy based on no pathophysiologic or epidemiologic sense. This is why we developed the Causal -Benefit model.1,5 Not to replace the Risk model. But to extend and amplify it. However, our model is not the model the authors present in this paper. The distinctions in purpose and method and outcomes are categorical. 

In our model, we calculate the minimum Absolute Risk Reduction (ARR) in those who would be selected for pharmacological therapy by the conventional risk model. We then identify those amongst the general population, who would have a similar ARR but are below the threshold level of risk to be eligible for pharmacological therapy. These are added to the initial risk group. By design, therefore, the population selected by the ARR approach must be larger than the conventional approach and the levels of LDL-C (or whatever the determinative marker selected is- preferably apoB). We found that "Among 2013 guideline-eligible individuals, the mean 10-year predicted risk was 13.9% (range, 7.5%-43.5%) and the expected ARR10 from statins was 4.8% (range, 2.3%-10.6%), whereas for the individualized statin benefit approach, the mean 10-year predicted risk was 10.9% (range, 3.7%-43.5%) and the expected ARR10 was 4.0% (range, 2.3%-10.6%). This corresponds to a group NNT10 of 21 (range, 9-44) and 25 (range, 9-44) for statin eligibility based on 2013 guidelines or individualized benefit-based approach, respectively." 4 Amongst those added in the lower risk but acceptable benefit group, the average age was less and a greater number were women. The LDL-C, of course was higher. 4

By contrast, the major findings in the present analysis are: "49.2 million (95%CI: 45.3-53.0) and 50.3 million (95%CI: 46.0-54.6) adults would be eligible for statins therapy under the risk reduction-based and absolute risk-based strategies, respectively. Among 58.9 million adults recommended for statin therapy by either strategy, 16.5% of individuals are uniquely identified by the absolute risk-based approach and 14.6% by the risk reduction-based approach, with only 68.9% overlap." 

Positive features do emerge in the present paper: the ARR group is younger, has higher levels of LDL-C, and includes more females. But because the authors have committed themselves to a strategy that prevents an equal number of events with both strategies, they throw away the potential for gain. What is the logic for capping gain? Treating more with moderate risk is cost-effective?6 The outcome the authors have chosen is to cap is benefit for the people they have chosen to serve, not extend it. What is the argument against multiplying the benefit of preventing the most common cause of death and disability of humankind? In our view, further work must be done. We should move beyond the 10-year horizon of risk. Earlier intervention will produce greater benefit.7 

I do not doubt the authors' good intentions. But the strategy they have chosen frustrates their good intentions. I do hope the authors reconsider the principles on which they have based their analysis. I cannot insist they change. But I do insist that their paper be revised so as to indicate why they reach one result, whereas we reach another, very different result. They have not applied our method and this is why they have not reproduced our results. 

I disagree with how they have constructed their analysis and I have attempted to set out why. But my criticisms are offered in a spirit of collegiality and good will.

Reviewer #3: General comments

The authors compare two strategies to determine statin eligibility in primary prevention: one traditional, based on the absolute calculated CVD risk, and the other based on the predicted absolute risk reduction with statin therapy. The authors claim with robust, convincing data that the second strategy is more efficient.

The manuscript is original and reports relevant results on statin eligibility in primary prevention. It is very well-written, high-quality text.

The methodology is complex, but a detailed explanation is provided in the supplement.

Specific comments

1. An important point that is not clear in the text is the percentage of the population that would be statin eligible by either strategy to prevent a similar number of events. I suggest that the authors mention this information, preferentially in both scenarios (treating high risk/high benefit and treating at least intermediate risk/at least moderate benefit).

2. More importantly, in my opinion, is the shift of statin-eligible individuals to younger persons with fewer risk factors but higher LDL-c with the risk reduction-based strategy, compared to the traditional approach using the absolute calculated risk. This was also shown in previous studies addressing this topic (doi 10.1161/CIRCULATIONAHA.115.018383 [ref 10] and doi 10.1016/j.amjcard.2018.02.011). Authors may choose to mention that the observed results coincide with previous reports as a way of reinforcing the validity of this relevant finding. 

3. In the Discussion, the authors state "Early intervention in younger adults is critical, as cumulative LDL-C exposure during young and middle adulthood is associated with increased CVD risk, and the benefits of LDL-C lowering are more substantial with prolonged treatment." As a suggestion, I understand that this sentence would be a great opportunity to cite a paper by Thanassoulis et al (doi 10.1001/jamacardio.2018.3476) showing data in favor of a long-term benefit-based strategy for statin eligibility in younger persons.

4. Results from analyses considering statin therapy using the moderate-benefit cut-off or for intermediate-risk group are shown in Table 1 and Figure 2, but are not mentioned in the main text. I suggest that the authors comment on these findings in the text.

5. The authors state that "The conceptual diagram and detailed methods for evaluating these outcomes are provided in the S1 Appendix", but the conceptual diagram is shown in Figure 1.

---

* Please upload any figures associated with your paper as individual TIF or EPS files with 300dpi resolution at resubmission; please read our figure guidelines for more information on our requirements: http://journals.plos.org/plosmedicine/s/figures. While revising your submission, please upload your figure files to the PACE digital diagnostic tool, https://pacev2.apexcovantage.com/. PACE helps ensure that figures meet PLOS requirements. To use PACE, you must first register as a user. Then, login and navigate to the UPLOAD tab, where you will find detailed instructions on how to use the tool. If you encounter any issues or have any questions when using PACE, please email us at PLOSMedicine@plos.org.

* FINANCIAL DISCLOSURES: The funding statement should include: specific grant numbers, initials of authors who received each award, URLs to sponsors’ websites. Also, please state whether any sponsors or funders (other than the named authors) played any role in study design, data collection and analysis, the decision to publish, or preparation of the manuscript. If they had no role in the research, include this sentence: “The funders had no role in study design, data collection and analysis, decision to publish, or preparation of the manuscript.”

* COMPETING INTEREST: All authors must declare their relevant competing interests per the PLOS policy, which can be seen here: https://journals.plos.org/plosmedicine/s/competing-interests

For authors with ties to industry, please indicate whether any of the interests has a financial stake in the results of the current study.

FIGURES AND TABLES

SUPPLEMENTARY MATERIAL

REFERENCES

* Where website addresses are cited, please include the complete URL and specify the date of access (e.g. [accessed: 12/06/2024]).

STUDY TYPE-SPECIFIC REQUESTS

* Abstract: Please include the study design, population and setting, number of participants, years during which the study took place (enrollment and follow up), length of follow up, and main outcome measures.

* Please ensure that the study is reported according to the STROBE (or appropriate STOBE extension) guideline (available from: https://www.equator-network.org/reporting-guidelines/strobe) and include the completed STROBE (or STROBE extension) checklist as Supporting Information. Please add the following statement, or similar, to the Methods: "This study is reported as per the Strengthening the Reporting of Observational Studies in Epidemiology (STROBE) guideline (S1 Checklist)." When completing the checklist, please use section and paragraph numbers, rather than page numbers. 

* For all observational studies, in the manuscript text, please indicate: (1) the specific hypotheses you intended to test, (2) the analytical methods by which you planned to test them, (3) the analyses you actually performed, and (4) when reported analyses differ from those that were planned, transparent explanations for differences that affect the reliability of the study's results. If a reported analysis was performed based on an interesting but unanticipated pattern in the data, please be clear that the analysis was data driven. 

* Please state in the Methods section whether the study had a prospective protocol or analysis plan. If a prospective analysis plan (from your funding proposal, IRB or other ethics committee submission, study protocol, or other planning document written before analyzing the data) was used in designing the study, please include the relevant document(s) with your revised manuscript as a Supporting Information file to be published alongside your study and cite it in the Methods section. A legend for this file should be included at the end of your manuscript. If no such document exists, please make sure that the Methods section transparently describes when analyses were planned, and when/why any data-driven changes to analyses took place. Changes in the analysis, including those made in response to peer review comments, should be identified as such in the Methods section of the paper, with rationale.

---

## [Decision Letter · Decision Letter 2]

Dear Dr. Gao,

Thank you very much for re-submitting your manuscript "Absolute risk-based versus individualized benefit approaches for determining statin eligibility in primary prevention of cardiovascular diseases in Chinese populations" (PMEDICINE-D-25-00419R2) for review by PLOS Medicine.

I appreciate the detailed response to the reviewers' and editors’ comments. I have discussed the paper with my colleagues, and it has also been seen again by all of the original reviewers. The changes made to the paper were mostly satisfactory to the reviewers. As such, we intend to accept the paper for publication, pending your attention to the reviewers' and editors' comments below in a further revision. When submitting your revised paper, please once again include a detailed point-by-point response to the editorial comments.

[LINK]

In revising the manuscript for further consideration here, please ensure you address the specific points made by each reviewer and the editors. In your rebuttal letter you should indicate your response to the reviewers' and editors' comments and the changes you have made in the manuscript. Please submit a clean version of the paper as the main article file. A version with changes marked must also be uploaded as a marked up manuscript file. Please also check the guidelines for revised papers at http://journals.plos.org/plosmedicine/s/revising-your-manuscript for any that apply to your paper.

We ask that you submit your revision within 1 week (June 13 2025). However, if this deadline is not feasible, please contact me by email, and we can discuss a suitable alternative.

Please do not hesitate to contact me directly with any questions (atosun@plos.org). 

We look forward to receiving the revised manuscript.   

Sincerely,

Alexandra Tosun, PhD

Associate Editor 

PLOS Medicine

plosmedicine.org

Comments from Reviewers:

Reviewer #1: Thank you for revising the paper based on my comments. I enjoy reading the discussion about statin pathways and the new density plots on iARR. 

I feel my comments are mostly addressed, even though I am still uncertain about the second line of the conclusion 'This suggests potentially improved efficiency in targeting individuals for statin therapy, which may be particularly relevant for shared decision-making in resource-constrained settings with distinct population risk profiles.' Efficiency arises from cost-effectiveness which has not been studied here. Given the population level effectiveness is similar I do not envision to see a massive increase in efficiency. Instead I would suggest to replace this line with 'The potential increase in population and individual level efficiency warrants further research'.

In the first paragraph of the discussion 'was more efficient at the population level' does not seem to be supported by the findings. I'd also advise similar revision/rewording in the Conclusion section in the main text. 

Reviewer #2: Thank you. This version is the product of a major additional effort. But the product is impressive. This analysis now substantially extends understanding of the relative advantages and disadvantages of the Causal-Benefit Model of Cardiovascular Prevention versus the conventional Risk Model of Cardiovascualar Prevention. 

I am not requesting further change. My only comment for the authors to consider going forward is the issue of whether the benefit from statin therapy is attributable, at lease in part, to pleiotropic effects. I am not persuaded this is the case. Plaque stabilization could well be a happy outcome of lowering apoB particle number. But whether this is the case or not is irrelevant to the analysis. The concept is anchored in the CCT finding that reduction in event rate is close to constant per mmol/l reduction in LDL-C. This is a mathematic relation not a proof of mechanism. 

Reviewer #3: The authors made appropriate changes to the manuscript and the responses to the comments were satisfactory.

No further comments.

[LINK]

Requests from Editors:

GENERAL

* Please confirm that your title complies with to PLOS Medicine's style. Your title must be nondeclarative and not a question. It should begin with main concept if possible. "Effect of" should be used only if causality can be inferred, i.e., for an RCT. Please place the study design ("A randomized controlled trial," "A retrospective study," "A modelling study," etc.) in the subtitle (i.e., after a colon).

* Statistical reporting: Please revise throughout the manuscript, including tables and figures.

- Please report statistical information as follows to improve clarity for the reader ""22% (95% CI [13,28]; p</=)"".

- Please separate upper and lower bounds with commas instead of hyphens as the latter can be confused with reporting of negative values.

- Please repeat statistical definitions (HR, CI etc.) for each set of parentheses.

* Please ensure that all abbreviations are defined at first use throughout the text (including statistical abbreviations). Please also check figures and tables.

* Please ensure that all tables and figures, including those in supplementary files, are appropriately referenced in the main text.

* The terms gender and sex are not interchangeable (as discussed in https://www.who.int/health-topics/gender#tab=tab_1 ); please use the appropriate term.

* Please include the statement on code availability in the data availability statement (in the online submission form).

* In your study causality cannot be inferred. Please remove language that implies causality, such as impact. Please use associational language instead.

ABSTRACT

* Please confirm that your abstract complies with our requirements, including providing all the information relevant to this study type https://journals.plos.org/plosmedicine/s/submission-guidelines#loc-abstract

* Please ensure that all numbers presented in the abstract are present and identical to numbers presented in the main manuscript text.

* When revising the Abstract, please include sufficient explanation and guidance throughout to ensure that the reader can understand and follow the synopsis of the results and conclusions.

* We're not sure if the reader will understand when you write, "A total of 7,287 adults were analyzed, representing 324.6 million Chinese adults." Suggestion: "A total of 7,287 adults were analyzed, forming a cohort reflective of 324.6 million Chinese adults."

* “The benefit approach uniquely identified 8.6 million, including 1.3 million with borderline risk (5%-7.5%).” – please explain in more detail. Suggestion: The benefit approach alone identified 8.6 million people who would highly benefit from statin therapy. This includes 1.3 million people with borderline risk (5%-7.5%), who would not be eligible for statin therapy under the absolute risk-based approach.

* “The risk-based approach selected more individuals with low predicted iARR (longer lower-end tail in the iARR distribution), indicating better individual level targeting efficiency with the benefit approach.” – We suggest including the numerical results here. Additionally, we believe the explanation from the rebuttal ("meaning it selects more individuals with low predicted benefit") would be useful.

AUTHOR SUMMARY

* “Applying the Causal-Benefit model's minimum benefit threshold principle, the benefit approach recommended treatment for significantly more adults (46 million additional)…” – please check whether the use of the statistical term ‘significant’ is supported by the data, and if not please remove it.

METHODS AND RESULTS 

* “blood-based biomarker collection study was also approved [14].” – please provide the IRB approval number and clarify that written informed consent was provided for both studies.

* l.58: When providing age, please ensure to include a unit, such as years. Please revise throughout the manuscript.

* Figure 2: Please clarify what the numbers in the circles represent and provide the denominator.

* l.264, we suggest adding a qualifier: “This demonstrates the potential superior efficiency of the individualized benefit approach in excluding individuals predicted to derive minimal absolute benefit from statin therapy.”

* Please check that any use of statistical terms (such as trend or significant) are supported by the data, and if not please remove them. 

* Please specify the variables controlled for in all relevant Tables.

* S2 Figure: Please note that the graphs say “Individualize”.

DISCUSSION

* Pleas remove all subheadings.

General Editorial Requests

---

## [Editor Report · Decision Letter 3]

Dear Dr Gao, 

On behalf of my colleagues and the Academic Editor, Andre P Kengne, I am pleased to inform you that we have agreed to publish your manuscript "Absolute risk-based versus individualized benefit approaches for determining statin eligibility in primary prevention of cardiovascular diseases in Chinese populations: A modelling study" (PMEDICINE-D-25-00419R3) in PLOS Medicine.

I appreciate your thorough responses to the reviewers' and editors' comments throughout the editorial process. We look forward to publishing your manuscript, and editorially there are only a few remaining points that should be addressed prior to publication. We will carefully check whether the changes have been made. If you have any questions or concerns regarding these final requests, please feel free to contact me at atosun@plos.org.

Please see below the minor points that we request you respond to:

1) Please note that the revised text does not contain the qualifier ‘potential’ as suggested (lines 300-302, track changes). Please change to: “This demonstrates the potential superior efficiency of the individualized benefit approach in excluding individuals predicted to derive minimal absolute benefit from statin therapy.”

2) Please remove the ‘Conclusion’ subheading. The final concluding paragraph should be a continuous part of the discussion.

3) We believe the TRIPOD checklist is more appropriate for your study. If you agree, please replace the STROBE checklist with a completed TRIPOD checklist (using sections and paragraphs), and update the manuscript accordingly. Also, include the following statement in the Methods section: "This study followed the Transparent Reporting of a Multivariable Prediction Model for Individual Prognosis or Diagnosis (TRIPOD) Statement [22]."

Before your manuscript can be formally accepted you will need to complete some formatting changes, which you will receive in a follow up email (including the editorial points above). Please be aware that it may take several days for you to receive this email; during this time no action is required by you. Once you have received these formatting requests, please note that your manuscript will not be scheduled for publication until you have made the required changes.

PRESS

Sincerely, 

Alexandra Tosun, PhD 

Senior Editor 

PLOS Medicine